# Forcing the reversibility of a mechanochemical reaction

Amy E.M. Beedle [1], Marc Mora[1], Colin T. Davis[2], Ambrosius P. Snijders [2], Guillaume Stirnemann[3] & Sergi Garcia-Manyes [1,4]

Mechanical force modifies the free-energy surface of chemical reactions, often enabling thermodynamically unfavoured reaction pathways. Most of our molecular understanding of force-induced reactivity is restricted to the irreversible homolytic scission of covalent bonds and ring-opening in polymer mechanophores. Whether mechanical force can by-pass thermodynamically locked reactivity in heterolytic bimolecular reactions and how this impacts the reaction reversibility remains poorly understood. Using single-molecule force-clamp spectroscopy, here we show that mechanical force promotes the thermodynamically disfavored $S_N2$ cleavage of an individual protein disulfide bond by poor nucleophilic organic thiols. Upon force removal, the transition from the resulting high-energy unstable mixed disulfide product back to the initial, low-energy disulfide bond reactant becomes suddenly spontaneous, rendering the reaction fully reversible. By rationally varying the nucleophilicity of a series of small thiols, we demonstrate how force-regulated chemical kinetics can be finely coupled with thermodynamics to predict and modulate the reversibility of bimolecular mechanochemical reactions.

[1] Department of Physics and Randall Centre for Cell and Molecular Biophysics, King's College London, London WC2R 2LS, UK. [2] The Francis Crick Institute, Protein analysis and Proteomics Science Technology Platform, 1 Midland Road, London NW1 1AT, UK. [3] CNRS Laboratoire de Biochimie Théorique, Institut de Biologie Physico-Chimique, Univ. Paris Denis Diderot, Sorbonne Paris Cité, PSL Research University, 13 rue Pierre et Marie Curie, 75005 Paris, France. [4] The Francis Crick Institute, 1 Midland Road, London NW1 1AT, UK. These authors contributed equally: Amy E.M. Beedle, Marc Mora. Correspondence and requests for materials should be addressed to S.G.-M. (email: sergi.garcia-manyes@kcl.ac.uk)

Chemical reactions typically require reactants to cross a free-energy barrier before transforming into products. Reaction rate theories predict that the reaction probability decreases exponentially with the free-energy barrier height ~exp $[-\Delta G^{\neq}/k_{B}T]$. Temperature, light, and electric current have been classically used to activate chemical reactions by overcoming such energy barrier. Albeit being far less studied, mechanical force has emerged as an orthogonal, fundamentally distinct way to activate chemical reactions[1–3], frequently biasing the free-energy surface towards thermodynamically unfavoured pathways[4]. For example, the application of ultrasounds in bulk experiments yield polymeric products that cannot be generated by conventional energy sources[5]. Alternatively, single-molecule experiments—which afford the advantage of localizing mechanical stress on a single chemical bond[6]—have enabled the mechanical activation of covalent bonds[7], ultimately resulting in irreversible homolytic[8–12] and heterolytic[13–15] bond scission, and the electrocyclic ring-opening of stress-responsive mechanophores[16–18], which often violate the orbital symmetry-conservation rules[19–24]. However, the precise mechanisms by which mechanical force can trigger thermodynamically-uphill reactivity in bimolecular heterolytic systems and how this impacts the reaction reversibility remains largely unknown.

In the most simple 1D energy diagram, a chemical reaction is non-favored if the free energy of the products is higher than that of the reactants ($\Delta G > 0$) and the activation barrier (defined here as the free-energy difference between the reaction transition state and the reactants) is significant. According to this picture, the application of force could potentially lower the activation barrier such that its peak becomes close to the free energy of the reactants, eventually enabling an otherwise unlikely transition. An immediate consequence is that, upon force removal, the opposite reaction, bringing the thermodynamically non-favored products back to the reactants, should become suddenly spontaneous. Despite its simplicity, this theoretical reactivity see-saw scenario has not been experimentally tested. Arguably, thiol-disulfide exchange is one of the reactions in organic chemistry that is best studied at the fundamental level, both experimentally[25] and from the theoretical[26,27] viewpoints. Pioneering single-molecule mechanochemical experiments demonstrated that mechanical force accelerates the rupture of an individual protein disulfide bond[15]. All these experiments employed thiol nucleophiles for which the disulfide bond cleavage reaction was thermodynamically favored[28–30]. Due to its mechanistic simplicity[31], the concerted thiol-disulfide nucleophilic substitution is an ideal case-study platform to interrogate whether mechanical force can also switch on the reactivity of thermodynamically stable reactants, for which the reaction is energetically non-spontaneous.

Here we show, using single-molecule force-clamp spectroscopy, that mechanical force triggers the otherwise inhibited bimolecular nucleophilic substitution ($S_N2$) cleavage of a protein disulfide bond by poor nucleophilic thiols, rendering the reaction unexpectedly reversible upon force withdrawal.

## Results

### Thermodynamically non-favored disulfide bond rupture.
To elucidate the spontaneous character of disulfide cleavage, we first chose a series of small thiols (Fig. 1a) which, based on their chemical structure and physicochemical properties[32,33], encompass a broad range of sulfur $pK_a$ values that we individually determined experimentally (Supplementary Figs. 1–9). We next conducted DFT calculations in implicit solvent using two different approaches (B3LYP/6-311 G + (d,p) or M06-2X/aug-cc-pVDZ with one explicit water, see Methods) to estimate the standard free-energy $\Delta G°$ associated with the reduction of the

disulfide bond present in cystine (the dimer of cysteine) by each small thiol (Supplementary Table 1). Since the inclusion of one explicit water molecule in the calculations using the M06-2X functional leads to a better agreement between calculated and experimental $pK_a$ values[33] (Supplementary Table 1) in the following we will refer to the corresponding free-energy values, but the conclusions remain qualitatively similar for B3LYP (Supplementary Fig. 10). The wide range of calculated $\Delta G°$ values predicts a largely different reactivity of each thiol (Fig. 1b); while cysteine-methyl-ester (Cys-ME) cannot spontaneously reduce the disulfide bond ($\Delta G° = +7.3$ kcal mol$^{-1}$), mesna and N-acetyl-cysteine (NAC) display the largest negative $\Delta G°$ values, underpinning their efficiency as reducing agents ($\Delta G° = -6.7$ and $= -9.6$ kcal mol$^{-1}$, respectively). As expected, cysteine displays $\Delta G° = 0$ kcal mol$^{-1}$, since it is in 50:50 equilibrium ratio with cystine. Similarly, penicillamine, which exhibits a lower $pK_a$ than cysteine, is marginally non-spontaneous. The rest of the studied compounds, namely NAC-methyl-ester, glutathione, thioglycerol, and 1-mercapto-2-propanol display intermediate and negative $\Delta G°$ values. The classical colorimetric Ellman's assay[34], which probes the tendency of each small thiol to reduce a disulfide bond in solution, agrees well with the theoretically predicted reactivity trend (Fig. 1c, Supplementary Fig. 11). Whereas good reducing agents (negative $\Delta G$ values) turned the measuring solution yellow, those associated with a positive $\Delta G$ value resulted in an unaltered, transparent solution, suggesting the absence of reactivity towards disulfide bond reduction. Absorbance quantification confirmed the vanishingly small change in color of penicillamine and especially Cys-ME when compared to the blank, and demonstrated that under experimental conditions the reaction is under thermodynamic control (Supplementary Fig. 11).

We next questioned whether the same reactivity trend was observed in a chemically complex environment of a protein whereby the disulfide bond is embedded within its 3D structure. To address this question, we conducted intact protein Mass Spectrometry (MS) experiments targeted to quantify the ability of a subset of small thiols to reduce a protein disulfide bond present in a mutant of the 27th Ig domain of titin (Fig. 1d, inset), which contains a buried disulfide bond between positions 24 and 55 that becomes solvent exposed upon acetonitrile denaturation (Supplementary Figs. 12–19). In qualitative agreement with the colorimetric Ellman's assay and DFT calculations, the MS experiments (Fig. 1d) conducted under high nucleophile concentrations (1:54 protein disulfide:active nucleophile) demonstrated that while NAC and mesna induced significant disulfide bond reduction (with ~80% and ~40% efficiency, respectively), cysteine and penicillamine showed a much lower reduction efficiency (~20–25%). Even under these high nucleophile concentrations where the reaction equilibrium is largely displaced towards the products, Cys-ME was not able to reduce the protein disulfide bond. By lowering the stoichiometric conditions (1:6), the signal of penicillamine became indistinguishable from the MS spectrum background, demonstrating that disulfide bond reduction by penicillamine is also unfavorable (Supplementary Fig. 20). Combined, these experiments demonstrated the large variation of reactivity induced by a series of chemically distinct small thiols, and confirmed that the stoichiometric disulfide bond reduction by Cys-ME (and, to a lower extent, by penicillamine) is thermodynamically impaired in solution (implying that the forward "reduction" reaction is slower than the reverse "oxidation" reaction) and also within the context of the protein.

### Force activates non-favored disulfide bond cleavage.
To test whether mechanical force is able to effectively activate the

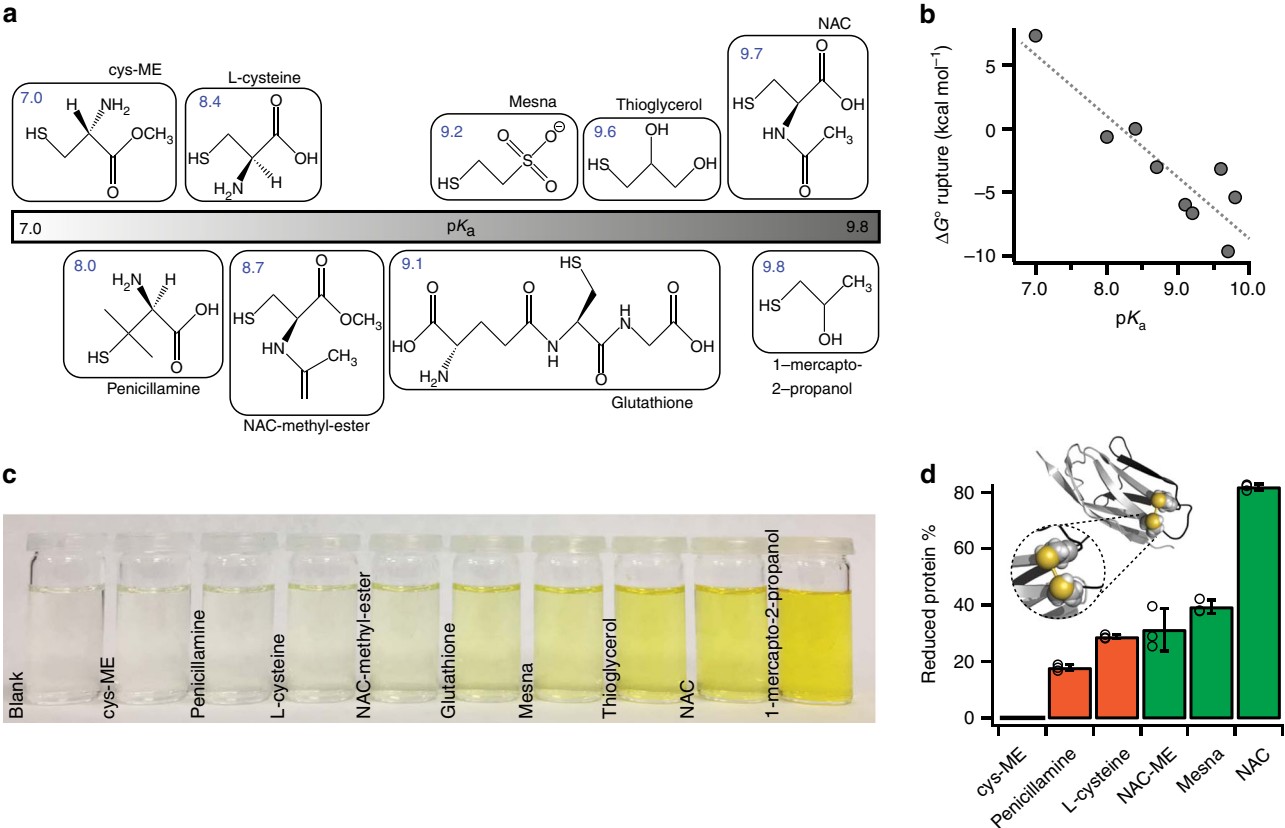

**Fig. 1** Thermodynamically unfavoured disulfide bond reduction. **a** The distinct physicochemical and reactivity properties of a number of small thiol-rich nucleophiles are exemplified in their largely different sulfur $pK_a$, spanning the range $pK_a$ ~7–10 (individual values represented on top of the chemical structures). **b** DFT calculations of the standard free energy (M06-2X functional with one explicit water) associated with disulfide rupture of cystine by the different thiol nucleophiles are in good agreement with the experimentally measured $pK_a$ values. **c** The Ellman's assay measures the reactivity of the small low molecular weight thiols in solution, displaying a reactivity trend that increases with the sulfur $pK_a$. **d** Mass spectrometry experiments measure the ability of each nucleophilic thiol to reduce an individual disulfide bond embedded within the core of a $I27_{E24C-K55C}$ protein (inset) under 1:54 (protein disulfide:active nucleophile) conditions upon acetonitrile exposure (green, thermodynamically allowed; orange, marginally allowed). The % of reduced disulfides matches the reactivity trend observed in solution and predicted by DFT calculations. (Empty circles represent values from individual experiments, $n = 3$, error bars: s.d)

thermodynamically unfavoured cleavage of a mechanically stretched protein disulfide bond by Cys-ME and penicillamine, we conducted a single-bond mechanochemical assay using force-clamp spectroscopy[15,28,30,35,36]. Briefly, a polyprotein made of eight identical repeats of the I27 mutant, $(I27_{E24C-K55C})_8$, was stretched between a gold-coated surface and an AFM cantilever tip (Fig. 2a) in the presence of 2 mM of deprotonated Cys-ME. The application of a constant force of 150 pN for 0.5 s triggered the mechanical unfolding of the protein (fingerprinted by steps of ~15 nm, gray regime) up to the structurally rigid disulfide bond, which becomes solvent exposed (Fig. 2b). The application of a second pulse at a higher force (350 pN) triggered disulfide bond reduction, marked by the presence of ~10 nm steps (red regime), which correspond to the extension of the amino acids that were trapped behind the disulfide bond. These experiments unambiguously demonstrate that mechanical force effectively promotes the thermodynamically unfavoured reduction of the I27 protein disulfide bond by Cys-ME (Fig. 2c). Similarly, penicillamine is also effective in triggering the mechanical reduction of the disulfide bond (Fig. 2d). Normalization and averaging a number of independent reduction trajectories at a constant force $F = 350$ pN allowed calculation of the reduction rate in the presence of each thiol compound at the same active (i.e., deprotonated, Supplementary Fig. 21) concentration (Fig. 2d). These experiments demonstrated that the reduction of the most thermodynamically

impaired Cys-ME is slower ($r = 0.40$ s$^{-1}$) than penicillamine ($r = 0.58$ s$^{-1}$). The reduction rates for both compounds are comparatively slower than that associated with the reduction by L-cysteine ($r = 2.48$ s$^{-1}$), for which $\Delta G° < kT$. Therefore, there is an apparent good correlation between thermodynamics and nucleophilicity—a kinetic phenomenon—generally following the rule of thumb stating that good nucleophiles are the conjugate bases of weak acids[37].

**Force-modulation of the reaction kinetics**. To compare the kinetics of disulfide reduction for all tested thiols (Fig. 1a) we measured their associated disulfide cleavage rate—normalized by their active (deprotonated) concentration—when stretched under $F = 350$ pN (Fig. 2e). The large spread in their associated reduction rates underscores their seemingly different nucleophilic efficiency. To rationalize these findings, we hypothesized that, at a given force, the effective charge of the electron lone pair of the attacking sulfur center might account for the distinct nucleophilicity of the tested compounds. Remarkably, the normalized rate of reduction of each small thiol at 350 pN scales with the electrostatic charge obtained by DFT calculations (Fig. 2e and Supplementary Table 1). We then set out to measure the force sensitivity of the reaction by comparing the force-dependency of the disulfide bond cleavage by different thiols. To address this

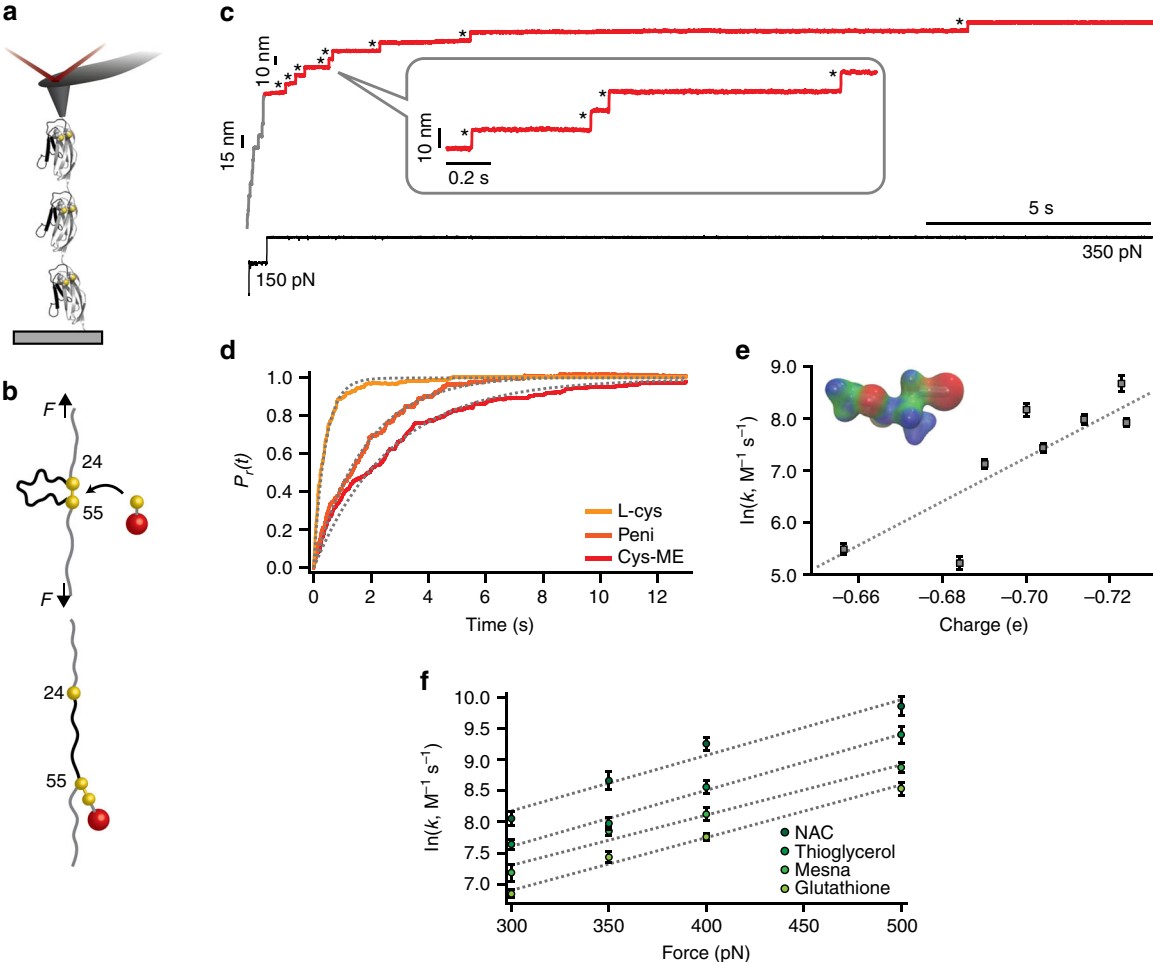

**Fig. 2** Mechanical force can induce thermodynamically non-spontaneous disulfide bond cleavage. **a** Schematic of the single-molecule force-clamp experiment, whereby a single ($I27_{E24C-K55C}$)$_8$ polyprotein is stretched by an AFM cantilever. **b** In the mechanochemical assay, mechanical force unfolds the protein (gray region) up to the mechanically rigid disulfide bond (formed between cysteine 24 and 55), which becomes solvent exposed. The presence of a nucleophile triggers disulfide cleavage, fingerprinted by the 10 nm steps that correspond to the extension of the protein amino acids that were trapped behind the disulfide bond (red region). The resulting reduced and mechanically unfolded protein harbors a mixed disulfide and a reduced protein thiolate. **c** Individual reduction trajectory demonstrating that the protein disulfide bond can be cleaved by 2 mM cysteine-methyl-ester (black asterisks mark the reduction of each individual disulfide bond under force, fingerprinted by a stepwise increase in the protein length of ~10 nm). **d** Disulfide bond rupture kinetics at $F = 350$ pN for the thermodynamically largely disfavored disulfide bond reduction by Cys-ME (red) and the marginally un-favored reaction in the presence of penicillamine (dark orange), compared to that of L-cysteine (orange). All compounds were kept at a constant deprotonated concentration [S$^-$] = 2 mM. **e** The concentration-normalized rate of rupture for all compounds at a constant $F = 350$ pN correlates with the electrostatic partial charge localized on the nucleophilic sulfur. (Inset: charge distribution plot of Cys-ME, from red (negative), to green (~0) to blue (positive)) (From left to right $n =$ 13, 24, 23, 17, 35, 23, 34, and 17, where $n$ = number of individual trajectories). **f** The force-dependent reduction kinetics for NAC, thioglycerol, mesna, and glutathione demonstrates that all compounds exhibit a similar force-sensitivity yet different nucleophilicity; (From left to right: $n_{NAC} = 35, 17, 42$, and 24; $n_{Thioglycerol} = 19, 35, 21$, and 27; $n_{Mesna} = 19, 35, 21$, and 27; $n_{Glutathione} = 37, 17, 41$, and 16; error bars: s.d)

question, we measured the reactivity of a selection of tested thiols across a wide range of stretching forces. We measured the rate of disulfide reduction for glutathione, mesna, thioglycerol, and NAC as a function of the pulling force spanning 300–500 pN (Fig. 2f). For each compound, a semi-log plot of the rate of reduction as a function of force displayed a straight line. Fitting of the Bell/Arrhenius equation [$r = r_0 \exp (F\Delta x/k_B T)$] normalized by the concentration to the experimental data yields the distance to the transition state ($\Delta x$) and the rate of the reduction in the absence of force ($r_0$)[15]. The normalized force-dependency of disulfide bond reduction for all tested compounds results in almost parallel curves, implying that the distance to the transition state remains practically invariant across all thiols ($\Delta x = 0.33$–$0.37$ Å). In agreement with previous findings[28], these results suggest that the chemical nature of the central atom is likely to set the reaction

sensitivity to force. By contrast, the height of the energy barrier, spanning 10.6 kT for NAC and 11.7 kT for glutathione assuming a constant pre-exponential value of $A = 10^7$ M$^{-1}$ s$^{-1}$ [38], is largely dependent on the different chemistry of each tested compound (Supplementary Table 2). Altogether, these results confirm that the partial charge of the attacking sulfur atom largely determines the height of the reaction-free-energy barrier, which can be decreased by the applied force with a similar scaling factor by all the tested thiols. Collectively, these experiments enable reconstruction of the 1D energy landscape of the force-activated reduction of the protein disulfide, whereby mechanical force triggers disulfide rupture in the presence of a variety of small thiol nucleophiles—even if the reaction is thermodynamically disfavored—to create a final protein structure involving a reduced protein thiolate and a mixed disulfide bond formed by the small

attacking nucleophile and the second reduced protein thiol (Fig. 2b). However, it remains unknown how the nature of the force-induced bimolecular nucleophilic attack by the chemically distinct thiol-rich reactants has knock-on effects on the reversibility of the reaction once the force is subsequently withdrawn, inducing the reformation of the initial disulfide bond.

**Mechanical regulation of the reversibility of the reaction.** To establish the probability of disulfide bond reformation we first need to consider the relative energy of the mixed disulfide formed in each case, which sets the tendency to undergo a nucleophilic attack by the neighboring and reduced protein thiolate if they are in close physical proximity (e.g., in the absence of force). For example, while the reduction of the disulfide bond by a good nucleophile such as mesna is thermodynamically favored ($\Delta G° = -6.7$ kcal mol$^{-1}$, Supplementary Table 1), the reverse reaction is energetically non-spontaneous, and hence the reformation of the initial disulfide bond will be disfavored. To directly confirm this hypothesis, we employed a five-pulse protocol that probes the extent of disulfide bond reformation (Fig. 3). As before (Fig. 2b) the initial pulse first unfolds the protein and subsequently triggers the mechanically activated cleavage of the disulfide bond, resulting in an extended and reduced protein form. After a quench pulse where the force is completely withdrawn ($F = 0$) for a period of time ($t_q = 8$ s), the test pulse probes the success of the disulfide bond reformation reaction. As expected, the low-energy mesna-mixed disulfide (Fig. 3a, green) is hardly attacked by the neighboring protein thiol, which is unable to displace mesna (a poor leaving group) and hence the reformation of the disulfide bond is largely hindered (<5%). This scenario is completely reversed when Cys-ME (red) is used as a nucleophile (Fig. 3b). As shown in Fig. 2c, mechanical force is able to induce the thermodynamically unfavoured reduction of the disulfide bond, also resulting in a related mixed disulfide product. But in this case the mixed disulfide is much higher in energy, and as such Cys-ME behaves as a good leaving group that can be readily attacked, in the absence of force, by the free protein thiol to spontaneously condense into the lower energy native disulfide bond reactant (Fig. 3b). In that sense, mechanical force is able to enhance the reversibility of a thermodynamically unfavoured reaction.

To establish a quantitative, experimental measurement of the degree of the reaction reversibility as a function of its associated free energy, we repeated the same force quench experiments in the presence of the complete series of small thiols and compared in each case the fraction of proteins that reformed the initial disulfide bond (Fig. 3c). Remarkably, we observed a linear relationship between the % of protein disulfide bond reformation and the experimental p$K_a$ (Fig. 3c) and also with the calculated standard free energy associated with disulfide rupture (Supplementary Fig. 22). While we have evidence of the presence of a kinetic activation barrier for the reverse reaction, the long quench time $t_q = 8$ s ensures that the reaction is under thermodynamic control, as further demonstrated for the case of NAC-methylester and 1-mercapto-2-propanol (Supplementary Fig. 23). We also examined whether the specific location of the disulfide bond within the structure of the I27 protein had a role in the disulfide bond reformation. To this purpose, we repeated the experiments reported in Fig. 3c with the analogous (I27$_{G32C-A75C}$)$_8$ polyprotein, in which the cysteine mutations were engineered in different positions of the I27 sequence. The same trend, albeit consistently shifted to a lower absolute value, was recapitulated with this construct (Supplementary Fig. 24). Together, these experiments highlight a good correlation between the thermodynamics of a mechanochemical reaction and the degree of reversibility.

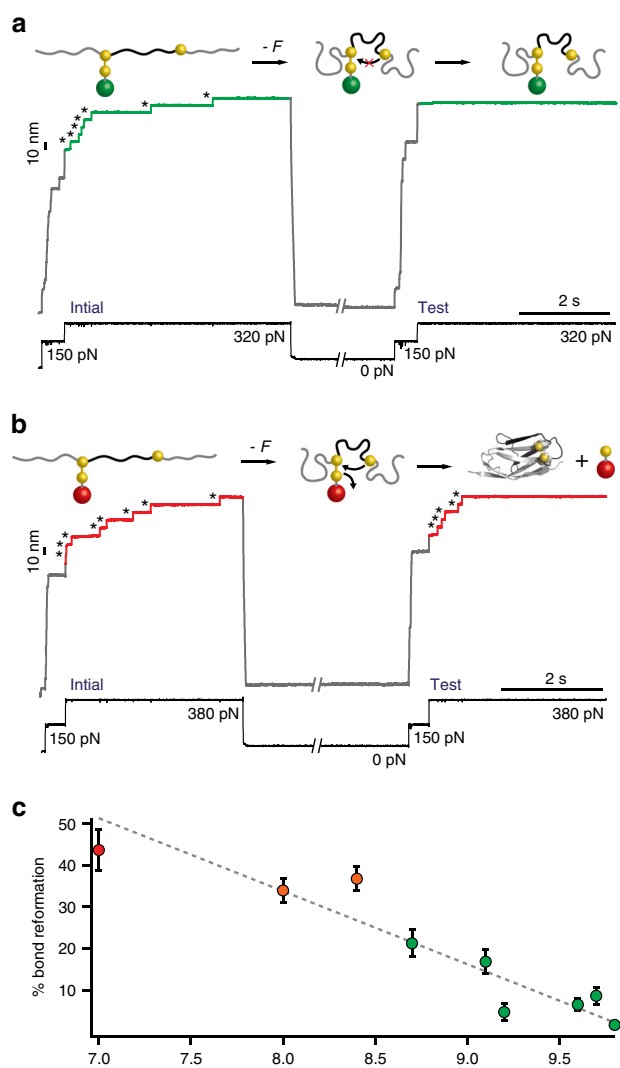

**Fig. 3** Mechanical activation of thermodynamically unfavoured reactions induces chemical reversibility. **a** Disulfide bond reduction by mesna ($\Delta G° = -6.7$ kcal mol$^{-1}$, green) results in a low-energy mixed disulfide that cannot reversibly reform the initial disulfide bond in the absence of force, marked by the absence of ~10 nm steps in the test pulse. **b** Conversely, reduction by Cys-ME ($\Delta G° = +7.3$ kcal mol$^{-1}$, red) gives rise to subsequent disulfide bond reformation, hallmarked by the ~10 nm stepwise increase in the test pulse. **c** The extent of disulfide bond reformation for all the tested compounds (green, thermodynamically allowed; orange, marginally allowed and red, thermodynamically unfavoured) exhibits a linear correlation with the p$K_a$ ($r^2 = 0.88$); (From left to right $n = 16, 35, 30, 24, 32, 33, 53, 31,$ and $39$; error bars: s.d)

## Discussion
Mechanical force has been shown to bias the shape of the underlying reaction energy surfaces, yielding products that are either prohibited or too slow to obtain when the reaction occurs under thermodynamic control[4,13,22]. Alternative studies have successfully followed the reformation of bonds upon force removal after mechanically induced scission[29,39–41]. However, we lacked quantitative evidence of the mechanisms controlling the reversibility of mechanochemical reactions that are thermodynamically non-spontaneous. Here we have employed single-molecule force-clamp spectroscopy to directly explore, with single-bond resolution, the reversibility of a mechanically triggered S$_N$2 reaction. Using a classical spectrometric assay, we first

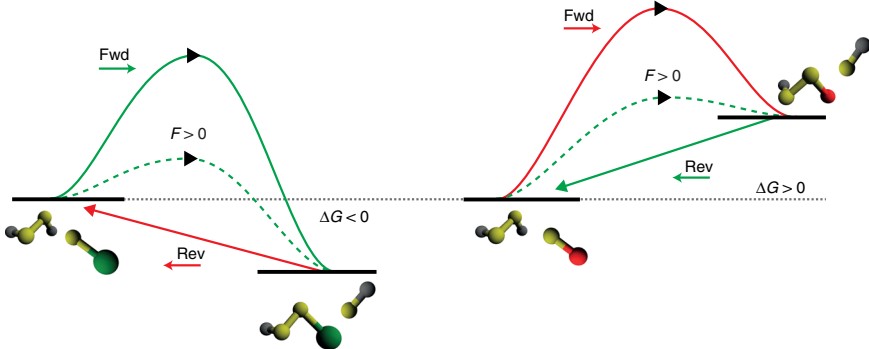

**Fig. 4** Mechanical force deforms the reaction energy landscape and modulates the reversibility of the reaction. Schematics of the 1D energy landscapes highlighting how the application of mechanical force to an individual protein disulfide in the presence of chemically distinct small thiolates induces disulfide bond rupture and governs the reaction reversibility. Using a thiol nucleophile that renders the disulfide bond cleavage reaction thermodynamically spontaneous ($\Delta G < 0$), mechanical force will lower the main energy barrier (dashed arrow), giving rise to a low free energy mixed-disulfide. Consequently, the reverse reaction (entailing the $S_N2$ reattack of the mixed disulfide bond by the neighboring protein reduced thiolate) will not be favorable, making the overall net reaction irreversible (red solid arrow). By contrast, mechanical force can trigger a thermodynamically un-favored reaction ($\Delta G > 0$) by also significantly lowering its activation energy (dashed arrow). In this case, the resulting mixed disulfide bond has higher energy than the initial reactants. Hence, upon force release, the reformation of the initial disulfide bond becomes suddenly spontaneous, rendering the overall reaction reversible (green solid arrow)

identified a series of small thiols exhibiting a large range of p$K_a$ values, which underpin their distinct nucleophilic reactivity. The standard free energy corresponding to the reduction of a simple disulfide bond by each compound was calculated using DFT, providing results in general agreement with the experimental p$K_a$ determination. Crucially, the relative reactivity observed for small thiols in solution is recapitulated when the disulfide bond is embedded within the complexity of the I27 protein environment, as revealed by MS measurements.

We next employed single-molecule force-clamp spectroscopy to study how mechanical force modulates the reactivity of individual protein disulfide bonds. As expected, the application of force triggered the acceleration of disulfide bond cleavage in the presence of reactive nucleophiles, i.e., those exhibiting $\Delta G < 0$. From the force-dependency of reduction, we obtained the relevant parameters defining the 1D projection of the energy landscape. In particular, we measured a distance to the transition state of $\Delta x \sim 0.35$ Å that is almost invariant for all tested S-nucleophiles, whereas the height of the energy barrier is largely dictated by the chemical structure, and notably by the partial electrostatic charge located on the nucleophilic sulfur. In this vein, mixed quantum-classical (QM/MM) calculations have suggested that mechanical force increases the redox potential of the disulfide bond[42], and therefore that a thermodynamic destabilization of the oxidized disulfide bond can influence the free-energy barrier. However, this possible contribution should remain constant in our experiments as it only relates to the disulfide bond reactant at a given force, and hence should not affect the experimental trend that we observed upon varying the different attacking nucleophiles.

With these considerations in mind, our experimental data allowed simple schematic reconstruction of the experimentally accessible 1D free-energy landscape (Fig. 4), which helped rationalize from a mechanistic perspective our unanticipated observations. In all cases for which thiol-mediated disulfide rupture is spontaneous, the force-activated reduction of the disulfide results in a low-energy mixed disulfide product (Fig. 4, green). Upon quenching the force, the reverse reaction, whereby the free protein thiol can re-attack the mixed disulfide to reform the initial disulfide bond, is uphill ($\Delta G > 0$), and as such the probability of disulfide bond reformation is vanishingly small (red solid arrow). Strikingly, our experiments unambiguously

demonstrate that nucleophiles triggering a non-spontaneous reaction can also cleave the protein disulfide if the activation barrier is sufficiently lowered by force (dashed green arrow) to be crossed on our experimental timescale window. However, in this case the mixed disulfide bond has a higher energy than the initial products, rendering the reverse reaction spontaneous ($\Delta G < 0$) hence enhancing disulfide bond reformation (Fig. 4, green solid arrow) after the pulling force is released.

Of note, the $\Delta G$ values obtained in our DFT calculations should only be considered as a simple guideline, as they do not take into account the chemical complexity of the protein structure nor the steric accessibility imposed by the protein fold. In fact, changing the location of the disulfide bond within the protein structure—namely between positions 32 and 75, (I27$_{G32C-A75C}$)$_8$—has a substantial effect on the overall efficiency of disulfide bond reformation (Supplementary Fig. 24), although the dependency with the reaction-free energy surprisingly recapitulated that observed for (I27$_{E24C-K55C}$)$_8$. Refined ab-initio molecular dynamics[43] or mixed quantum-classical simulations under force that take into account the protein surrounding[44,45], which are beyond the scope of this work, should certainly provide more accurate descriptions. However, despite the simplicity of our approach, the obtained $\Delta G°$ values provided here are sufficient to recover the experimental reformation trend observed in Fig. 3c.

Given that the studied disulfide bond is embedded within the protein structure, its rupture and reformation dynamics has implications on the broader context of oxidative protein folding[46]. In fact, disulfide bonds play an important role in maintaining the structure of proteins, both by providing enthalpic stabilization of the native state and also by creating signaling switches in response to oxidative stress[47,48]. From the mechanical perspective, disulfide bonds are covalent staples that largely regulate protein elasticity[49]. Therefore, control of the disulfide bond dynamics under force is a functionally inherent mechanism for proteins that are physiologically exposed to mechanical stress. In general, in-vivo reformation of disulfide bonds is typically catalyzed by dedicated enzymes[50,51], often employing the low p$K_a$ value of the catalytic cysteine as a general mechanism to foster reactivity[52]. However, other non-enzymatic oxidative pathways are slowly coming to light, involving sulfenic acid[40,53] or small thiols[54,55], such as the ones studied here. In this vein, our

experiments revealed that reducing the disulfide bonds by poor nucleophiles leads to unstable and reactive mixed disulfide products that can readily condense with the neighboring protein thiolate to form a stiff disulfide bond, yielding a properly refolded and mechanically rigid protein form. By contrast, disulfide cleavage by good nucleophiles gives rise to stable mixed disulfides, and their reactivity has important nanomechanical implications; if the energetically favorable mixed disulfide is unreactive, it might not be able to reform the initial disulfide bond, ultimately resulting in a misfolded protein devoid of mechanical stability. Alternatively, the solvent-exposed mixed disulfide bond can react with free thiols in solution (Supplementary Fig. 25), yielding a fully reduced protein that can refold into a native conformation that exhibits intermediate mechanical properties between the compliant, misfolded protein and the stiff, fully oxidized form[54]. The proportion of refolded yet reduced protein experimentally measured in the presence of each distinct thiol nucleophile follows surprisingly well thermodynamic considerations based on DFT calculations of the mixed disulfide and the thiol/cysteine homodimers (Methods, and Supplementary Fig. 25).

Collectively, the experiments reported here provide a direct platform to rationalize and predict the degree of disulfide bond rupture and reformation by combining thermodynamic and force-induced kinetic considerations. On a broader context, our results complement recent studies, mostly restricted to synthetic polymers, demonstrating that mechanical force can steer chemical reactions into products that are not generally sampled under thermodynamic control. Critically, here we add the notion of reversibility to the study of bimolecular and heterolytic mechanochemical reactions, suggesting that mechanical force can act as a reactivity lever able to unlock a chemical reaction and render it fully reversible. This reaction bi-directionality has larger nanomechanical implications if the reaction occurs within the core of a protein with a physiological elastic role such as cardiac titin.

## Methods

**Protein engineering.** The (I27$_{E24C-K55C}$)$_8$ and (I27$_{G32C-A75C}$)$_8$ polyproteins were subcloned using the BamHI, BglII, and KpnI restriction sites. Both the polyprotein constructs and the I27$_{E24C-K55C}$ monomer were cloned into the pQE80L (Qiagen) expression vector, and transformed into the BLR(DE3) *Escherichia coli* expression strain. Cells were grown at 37 °C in LB broth supplemented with 100 μg/mL ampicillin. Upon an OD$_{600}$ of ~0.6, Isopropyl β-D-1-thiogalactopyranoside (1 mM) was used to induce the cultures and they were then incubated overnight at 20 °C. The cells were harvested and then disrupted using a French Press. The proteins from the lysate were double purified, first by metal affinity chromatography on Talon resin (Clontech) and then by gel-filtration using a Superdex 200 10/300 GL column (GE Biosciences). Following each purification the protein concentration was estimated using the Bradford protein assay. Protein sequences provided in Supplementary Table 3.

**Colorimetric Ellman's assay.** Classic colorimetric Ellman's[34] assays were conducted by incubating freshly prepared solutions of the tested small thiols with 5,5′-dithio-bis-(2-nitrobenzoic acid) (DTNB). DTNB contains a symmetric disulfide bridge that can be reduced by the different small thiols, yielding to two different products, a mixed disulfide and 2-nitro-5-thiobenzoic acid (TNB). TNB is responsible for the characteristic yellow color of Ellman's solutions, serving as an indicator of disulfide bond reduction. DTNB was prepared at a final concentration of 1 mM with ~3 ml of ethanol at pH = 7.5 in PBS buffer. Solutions of all nine different small thiols were prepared at a final deprotonated sulfur concentration of 0.5 μM. The blank solution was prepared by mixing 1:1 DNTB solution in PBS buffer. Final solutions of small thiols and DNTB were prepared by mixing them in vials at a 1:1 ratio. All reactions were performed at room temperature. Absorbance measurements were conducted using a JENWAY 6305 spectrophotometer (at λ = 412 nm) on the same solutions contained in the aforementioned vials.

**UV spectrophotometric measurements for pK$_a$ determination.** Determination of the sulfur pK$_a$ values for the studied thiols was conducted through absorbance measurements using a JENWAY 6305 spectrophotometer at λ = 240 nm[56]. Solutions were prepared at a final concentration of 250 μM of the tested thiols in PBS buffer. Prior to each absorbance reading, the pH of the solution was measured with

a Mettler Toledo pH-meter. Absorbance measurements were conducted from acidic to basic pH values, whereby pH increase was achieved by successive addition of small aliquots of sodium hydroxide. The sulfur pK$_a$ values were obtained by fitting the Henderson-Hasselbach equation to the absorbance-pH data. Determination of the pK$_a$ values for L-cysteine, NAC, and 1-mercapto-2-propanol was performed with the addition of 1 mM of tris(2-carboxyethyl)phosphine (Sigma-Aldrich) to prevent dimerization.

**Intact protein MS experiments.** The I27$_{E24C-K55C}$ monomer samples were prepared by diluting 1 μl of protein (~170 μM) in 15 μl of acetonitrile and incubated for 15 min to trigger protein unfolding and concomitant disulfide bridge exposure to the solvent. Then, 1 μl of PBS (pH 7.5) was added containing Cys-ME, penicillamine, L-cysteine, NAC-methyl-ester, mesna, or NAC at the same concentration as the single-molecule AFM experiments and incubated for 5 min.

Immediately after protein incubation, the molecular mass values corresponding to the reduced and oxidized I27$_{E24C-K55C}$ forms were determined by using a microTOFQ electrospray mass spectrometer (Bruker Daltonics, Coventry, UK). The monomer was desalted using a 2 mm × 10 mm guard column (Upchurch Scientific, Oak Harbor WA) packed with Poros R2 resin (Perseptice Biosystems, Framingham).

A slight acidic solution (0.10% of acetic acid) and 10% acetonitrile was used to inject the protein into the column, after washing with the same acidic solvent the protein was eluted in 60% acetonitrile and 0.1% acetic acid. The desalted monomer was then injected to the mass spectrometer using an electrospray voltage of 4.5 Kv at a constant velocity of 3 μl/min. Maximum entropy software was used to deconvolute the mass spectra (Bruker Daltonics, Coventry, UK).

The reduced I27$_{E24C-K55C}$ was validated by checking the characteristic mass peak of each specific post-translational modification. Percentages corresponding to the reduced protein were calculated by averaging the ratio between the oxidized and reduced peak intensities after three different repeats.

**DFT calculations.** Electronic structure calculations using the density functional theory (DFT) were performed with the Gaussian 09 software[57]. Instead of a thermodynamic cycle, we used the simpler "direct method", which was shown to lead to successful results for a series of thiol compounds[33]: all the structures were minimized directly in an implicit water solvent. Frequency calculations were performed on these optimized structures. The reported standard free energies contain contributions from the energy at 0 K, the zero-point energy, thermal corrections at 298 K, and the solvation free energy at 298 K. The charges on the thiolate sulfur atom were determined using the Natural Bond Orbital analysis implemented in Gaussian 09[57]. For each thiol compound, we estimated the energies of four different species: the thiol, the thiolate, the heterodimer containing a mixed disulfide with L-cysteine, and the homodimer containing two monomers linked by a disulfide bond. The first two calculations allow to estimate the properties of the nucleophile species (considered to be the thiolate), and to estimate the pK$_a$; the heterodimer is a proxy for a mixed disulfide bond between the nucleophile and a protein cysteine; the homodimer is the product of a second nucleophilic substitution on a mixed disulfide.

We compared two different combinations of DFT functionals, basis sets and solvent descriptions. We first performed calculations using the B3LYP functional[58,59], the 6-311 G + (d,p) basis set[60], and the PCM model for the implicit water solvent[61]. However, it is known that explicitly taking into account explicit water molecules improves the agreement between calculated and experimental pK$_a$ values on these species[33]. We therefore repeated our calculations with one explicit water molecule donating an hydrogen-bond to the sulfur atom in the thiolate and thiol species (since the charges of sulfur atoms in disulfide bonds are very small, adding explicit water molecules in the dimers is not expected to improve the results and turned out to be challenging to converge, as the corresponding hydrogen-bonds are very unlikely to form). We used a functional that was shown to perform well in these conditions, M06-2X[62], in combination with the implicit SMD model for the rest of the water solvent[63], and the more extended and diffuse aug-cc-pVDZ basis set[64]. As expected, this strategy gave results in overall better agreement with experiments.

In all cases, we considered the most stable forms obtained after the calculations. For the carboxyl and amine groups, we considered the predominant forms around neutral pH, i.e., the carboxylate RCOO$^-$ and the ammonium RNH$_3^+$. Compounds containing both groups thus exist as zwitterions in the experiments around neutral pH, but sometimes, the neutral form RCOOH/RNH$_2$ was more stable in the calculations. With the M06-2X functional and explicit water, all species containing such a zwitterion were found to be the most stable states. With the B3LYP function in purely explicit solvent, only the thiol/thiolates of penicillamine and L-cysteine were more stable as zwitterions; all the heterodimer species were found to be slightly more stable (usually, by a few kcal mol$^{-1}$ at most) in their neutral state.

**Force-clamp spectroscopy.** Single-molecule force-clamp spectroscopy atomic force microscopy (AFM) experiments were conducted using both a home-made set-up[65] and a commercial Luigs and Neumann force spectrometer[66] at room temperature. Approximately 0.5–2 μL of protein in PBS solution (at a concentration of 1–5 mg ml$^{-1}$) was deposited onto a freshly evaporated and plasma cleaned

gold cover slide. Prior to each experiment, the cantilever ($Si_3N_4$ Bruker MLCT-AUHW) was calibrated using the equipartition theorem, giving a typical spring constant of ~12–17 pN nm$^{-1}$. Each protein was pulled by initially pushing the cantilever onto the surface (1000–2000 pN for ~1 s) to promote the adhesion between the protein and the cantilever. The piezoelectric actuator was then retracted to produce a set deflection (force), which was held constant throughout the entirety of the trajectory. An external, active feedback mechanism maintained constant force while the protein extension was recorded. The force feedback is based on a proportional, integral and differential amplifier (PID) whose output was fed to the piezoelectric positioner. In every experiment, the feedback response is limited to ~1–3 ms. All force traces were filtered using a pole Bessel filter at 1 kHz.

In the disulfide bond rupture kinetics experiments, the protein was first unfolded at 150 pN for 0.5 s. The force was subsequently raised up to 350 pN and left constant for a large, variable period of time to capture the full kinetics of disulfide reduction for each studied nucleophile. The deprotonated concentration of each nucleophile is detailed in Supplementary Figs. 1–9. The rupture kinetic experiments conducted with cysteine-methy-ester, penicillamine, and L-cysteine were limited to ~3 h due to time-dependent dimerization, supplementary Figs. 26-28).

In the disulfide bond reformation experiments, the protein was unfolded at 150 pN for 0.5 s, followed by a high force pulse (varying between 200–500 pN depending on the use nucleophile) for 5 s (Cys-ME experiments were left extended for 3 s to compensate for the slightly higher active concentration). The quench pulse where the pulling force is completely removed was set in all cases to $t_q = 8$ s. Experiments were performed keeping the deprotonated concentration of all nucleophiles constant to 1 mM at pH 7.5 (except for cysteine-methyl-ester, which was performed at 2 mM due to issues with single-molecule detachment). Each of the thiol nucleophiles (cysteine-methyl-ester (Sigma-Aldrich, 98%), penicillamine (Sigma-Aldrich, 98–101%), cysteine (Santa Cruz biotechnology, 98%), N-acetyl-cysteine-methyl-ester (Sigma-Aldrich, 90%), reduced glutathione (Sigma-Aldrich, 98%), sodium 2-mercaptoethanesulfonate (mesna) (Sigma-Aldrich, 98%), 1-thioglycerol (Sigma-Aldrich, 97%), N-acetyl-cysteine-methyl-ester (Sigma-Aldrich, 99%), and 1-mercapto-2-propanol (Sigma-Aldrich, 95%)) were prepared in sodium phosphate buffer (50 mM sodium phosphate ($Na_2HPO_4$ and $NaH_2PO_4$), 150 mM NaCl), and the pH was adjusted by adding NaOH (2 M). Each solution was filtered through a 0.2-μm membrane prior to each experiment. The solution was prepared fresh each day.

**Data analysis**. All data were recorded and analyzed using custom software written in Igor Pro 6.0 (Wavemetrics). For the disulfide kinetics studies, only recordings showing the signature of at least 5 unfolding events (~15 nm steps) followed by 5 reduction events (~10 nm steps) were analyzed. No traces that featured more than eight unfolding or reduction events were considered. No traces that featured an unfolding event (~15 nm) in the high force pulse were included in the analysis. To obtain the reduction rate, we summed and normalized the reduction trajectories at each particular force. We fitted the resulting summed trace with a single exponential to obtain the rate of reduction. To obtain the error at a given force, n traces were randomly selected from the original data set using the bootstrap method. These traces were summed and fitted to obtain a rate constant. This was repeated 500 times, giving a spread for the rate of reduction, which resulted in the standard deviation for the reduction rate.

For the disulfide reformation experiments, only trajectories with a minimum of five, and maximum of eight rupture events in the first high force pulse were considered. Furthermore, trajectories were only analyzed if the protein was extended to the same length in the initial and the probe pulses. Standard deviation for the refolding fraction was estimated through the bootstrap method[67], where each recording was treated as an independent data point.

**Code availability**. Code is available upon request.

**Data availability**. Data supporting this research, including the single-molecule nanomechanics experiments, can be obtained from the corresponding author upon reasonable request.

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

## Acknowledgements

We thank Dr. Palma Rico and Ms. Stephanie Board for help in protein purification. We wish to thank Steven Howell for his help with the top-down mass spectrometry experiment. A.E.M.B. is funded by an EPSRC DTP fellowship. M.M. is funded by a Fight for Sight PhD studentship. C.T.D and A.P.S are supported by the Francis Crick Institute, which receives its core funding from Cancer Research UK (FC001999), the UK Medical Research Council (FC001999), and the Wellcome Trust (FC001999). G.S. acknowledges support from CNRS through a PICS allocation (PICS07571) and from the "Initiative d'Excellence" program from the French State (Grant "DYNAMO", ANR-11-LABX-0011-01). This work was supported by the BHF grant (PG/13/50/30426), Royal Society Research grant (RG120038), the European Commission (grant agreement SEP-210342844), EPSRC Fellowship K00641X/1 and by the Leverhulme Trust Research Leadership Award (RL-2016-015), all to S.G.-M.

## Author contributions

A.E.M.B and S.G.-M. conceived research and designed experiments. A.E.M.B. and M.M. conducted single-molecule mechanical experiments and analyzed data. C.T.D. M.M. and A.P.S. performed protein-based mass spectrometry experiments and analyzed data; M.M. performed small compound mass spectrometry experiments. G.S. performed and ana-lyzed DFT calculations. S.G.-M. wrote the paper. All authors contributed in revising and editing the manuscript.

## Additional information

**Competing interests:** The authors declare no competing interests.

