## [Peer Review File · Nature Communications]

Reviewers' comments:

Reviewer #1 (Remarks to the Author):

The manuscript presented by Beedle et al. deals with a mechanochemical reaction which is studied using different included the DFT modelling. As a computational chemists, I evaluate the quality of the DFT computations.

As already pointed out by the previous referee, the quality of the DFT computations is satisfactory as long as the purpose is to give to the readers a qualitative explanations of some aspect fo the experiments.

The authors did not consider to include BSSE and Grimme dft-d3 model as was suggested. It is clear that this would be a more complete way to carry out these DFT computations. However it is also clear that in this case the effect would certainly have been small. Fior this reason I evaluate positively the level of the computations considered.

Reviewer #2 (Remarks to the Author):

The authored have addressed my questions adequately. I am happy to recommend its publication in Nature Communications.

Reviewer #3 (Remarks to the Author):

Let me be clear at the outset that I believe this work should ultimately be published, and that Nature Commun. is a very good venue for it. But I still believe that the authors are in some cases mis-stating and in other cases over-stating the results of their study, to an extent that is not necessary. This is a very excellent study of force-coupled disulfide exchange, and expanding to reactions that are thermodynamically uphill in the absence of force but thermodynamically downhill when coupled to force is a genuine contribution. In some cases the authors might be using language that is common in the field. It is not my place to review the prior literature. But given the focus of this paper, I think it is the authors' best interests to be perhaps more precise and specific than others in the field might have been.

I have organized my comments according to the numbering in the authors' response letter.

(1) Regarding the "catalyst" discussion: Not only is force not a substance, if one is going to take the stance that it merely modifies the activation barrier (it more appropriately contributes mechanical potential energy that is coupled to the chemical potential energy of the system, but let's go with the authors' framework) then one must also take the stance that in this case it modifies the Gibbs energy change of the reaction. In fact, the authors make this a central point of their paper! A catalyst does not change the final state of the system, only the rate to get there. Here, force fundamentally changes where the reaction ends up. One might say that the key conclusion of the paper is that force is very much NOT acting as a catalyst in this case. That doesn't mean it's any less interesting or important in my view, it's just that the word catalyst has its meaning and in the context of a journal bearing that term I thought it important to stick to the literal definition.

(2) About the extrapolation to zero force and time scales: Here's my point, and I don't think the

authors and I agree disagree as much as it seems, but there is a very important subtlety that needs to be addressed. There are a few ways to look at it. Let's consider the case of a protein that wants to be folded/formed (internal disulfide intact) at equilibrium. There is a rate for unfolding/disulfide scission, and a rate for disulfide forming/refolding (the reverse reaction). Since equilibrium lies to "folded," the rate of force-free "refolding/disulfide forming" must be faster than the rate of force-free "unfolding/disulfide scission." We conclude from the authors' data that the rate of the folding is on the order of several seconds, so the rate of unfolding must be much slower than that. But when the force-coupled kinetics are extrapolated to zero force, the apparent rate constant is much faster. So the mechanism being probed is not the rate determining step for protein unfolding; there is change in rate determining step.

I agree with the authors, it is likely that protein conformational changes are the r.d.s. at zero force, and disulfide scission the r.d.s. at higher force. But given that the r.d.s. changes, we don't really know a priori where the mechanism changes. This makes the entire extrapolation exercise itself potentially problematic.

The fact that it is a bimolecular reaction does not matter. The reaction being extrapolated is bimolecular (and I think the authors mean "second order" and not "first order" kinetics in their response), and so the force-free rate is also the bimolecular rate, even if under pseudo-first order conditions.

(3) On "thermodynamically allowed": I think my point was misconstrued. It is just that a high energy barrier and positive delta G do not mean that the forward reaction does not occur on a microscopic level. It just means that the rate might be slow, and especially that even if the forward reaction is fast, the reverse reaction is faster. This kind of language is often muddled except when someone is teaching thermodynamics, but again I think it is important that the authors be precise in their language here. Especially given the extrapolated kinetics mentioned in point (2), above, where the apparent force-free forward reaction has a time scale of milliseconds (or seconds, depending on the units in figure 2f, which should be added). To say that a reaction with a measurable (or easily extrapolated but substantial) rate constant is "impossible" or "forbidden" seems misleading, when "slower than the reverse reaction" is precise. Again, this doesn't make the work any less interesting or important, at least in my view, but seems quite appropriate given the nature of the work.

Especially because the thermodynamics of the reaction are so central to this paper, it would also be useful to be careful about using delta G (which considers the actual position Q of the system) vs. delta G naught (standard free energy of reaction vs. free energy of reaction), which have very different meanings but are mixed in places here that, if taken literally, misrepresent the actual thermodynamics at play.

(4) On "first bimolecular heterolytic uphill etc... reaction": I don't think the work is any less interesting or important if it's not truly the "first" of this type. And I don't think it is, unless you go on to qualify it further. The 2004 Sijbesma paper with coordination bonds was both heterolytic and thermodynamically uphill. Given it involves platinum complexes, it was likely associative and bimolecular, although the mechanism was not proven. There is stronger evidence in the 2006 Craig paper that the reaction is bimolecular; it is certainly also heterolytic and thermodynamically uphill. In both cases, the bonds were reformed once force was removed, as in this paper. Neither was studied to nearly the extent done here, however, but they absolutely demonstrated that force was necessary to drive an uphill exchange reaction forward (on both kinetic and thermodynamic fronts). This detracts nothing from the current paper!

(5) Heterolytic: My apologies – I had "heterolytic scission" of the type $A-B \rightarrow A^+ + B^-$ in my head, but the authors are quite right that they do not use that language.

This is excellent work, certainly publishable in Nature Commun. once the language and discussion is cleaned up a little further. Because the authors are diving into such rich thermodynamic discussion (which I am delighted to see), I just think that they have put themselves in territory where attention to detail in language is all the more important. No additional experiments are necessary, in my opinion.

Reviewer #4 (Remarks to the Author):

In this manuscript, Beedle et al. study the force-dependence of protein disulfide cleavage by a range of reducing agents. This chemistry is prohibitively challenging to study in the absence of force; disulfide cleavage is slow for weak reducing agents and there is insufficient product to adequately study the reverse reaction. The authors' did not further develop the force clamp methodology, but executed it thoroughly and masterfully. The supplementary experiments and calculations add a great deal of evidence in support of the single molecule experimental results. The breadth of reducing agents studies allows for correlations to be made that provide new insight into this important chemical reaction.

I believe this manuscript is a great fit for Nature Communications. I have gone in detail through the responses to all four reviewers. I was pleased to see that there was a DFT expert chosen among the reviewers who could provide the proper perspective on the calculations. I am satisfied that all points I raised in the earlier review were well-addressed.

We thank the reviewer for their insightful comments, which have truly improved the quality of our paper. Below, please find a point-point, detailed response to the issues raised by the reviewer (*verbatim* in italics).

Response to Reviewer #3

Let me be clear at the outset that I believe this work should ultimately be published, and that Nature Commun. is a very good venue for it. But I still believe that the authors are in some cases mis-stating and in other cases over-stating the results of their study, to an extent that is not necessary. This is a very excellent study of force-coupled disulfide exchange, and expanding to reactions that are thermodynamically uphill in the absence of force but thermodynamically downhill when coupled to force is a genuine contribution. In some cases the authors might be using language that is common in the field. It is not my place to review the prior literature. But given the focus of this paper, I think it is the authors' best interests to be perhaps more precise and specific than others in the field might have been. I have organized my comments according to the numbering in the authors' response letter.

R: We thank the reviewer for his/her very flattering and encouraging comments, and for being so thorough in the points s/he raises.

1. - Regarding the "catalyst" discussion: Not only is force not a substance, if one is going to take the stance that it merely modifies the activation barrier (it more appropriately contributes mechanical potential energy that is coupled to the chemical potential energy of the system, but let's go with the authors' framework) then one must also take the stance that in this case it modifies the Gibbs energy change of the reaction. In fact, the authors make this a central point of their paper! A catalyst does not change the final state of the system, only the rate to get there. Here, force fundamentally changes where the reaction ends up. One might say that the key conclusion of the paper is that force is very much NOT acting as a catalyst in this case. That doesn't mean it's any less interesting or important in my view, it's just that the word catalyst has its meaning and in the context of a journal bearing that term I thought it important to stick to the literal definition.

R: We definitely see that attributing force the role of a catalyst is not straightforward, at least in the literal and classical definition. We believe that there is still a bit of misunderstanding in the way we perceive the role that force plays in the reaction (although we don't think we disagree as much as it seems with the reviewer in fact). Our point is that force does not change the ΔG of the reaction, and it only modifies (lowers) the activation energy barrier (e.g. Wiita et al, PNAS 2006; Garcia-Manyes, Nat Chem 2009; Dopieralski et al. Nat Chem 2016, Dudko et al., PNAS 2008; Evans, Bioph J, 1997; Liang, JACS 2011; Ainaravapu, JACS 2008). The consequence, however, is that force changes where the reaction ends up (merely because without force the barrier would be too slow to appreciate a significant amount of product). One point that we want to re-emphasize is that in our reaction, while the reactant (the disulfide bond) is under force, the product (the mixed disulfide) is not, and hence makes the whole interpretation of the role of force a bit less obvious. Be that as it may, we fully agree that force might not fall within the classical definition of catalyst, however, this is not a central point anymore given the change in venue. In any case, we have changed the word *catalyst* and *catalyses* in the abstract to read '*Mechanical force modifies the free-energy surface of reactions*',

and also in the introduction '*mechanical force triggers the*', and throughout the text to remove any possible source of confusion.

2. - About the extrapolation to zero force and time scales: Here's my point, and I don't think the authors and I agree disagree as much as it seems, but there is a very important subtlety that needs to be addressed. There are a few ways to look at it. Let's consider the case of a protein that wants to be folded/formed (internal disulfide intact) at equilibrium. There is a rate for unfolding/disulfide scission, and a rate for disulfide forming/refolding (the reverse reaction). Since equilibrium lies to "folded," the rate of force-free "refolding/disulfide forming" must be faster than the rate of force-free "unfolding/disulfide scission." We conclude from the authors' data that the rate of the folding is on the order of several seconds, so the rate of unfolding must be much slower than that. But when the force-coupled kinetics are extrapolated to zero force, the apparent rate constant is much faster. So the mechanism being probed is not the rate-determining step for protein unfolding; there is change in rate determining step. I agree with the authors, it is likely that protein conformational changes are the r.d.s. at zero force, and disulfide scission the r.d.s. at higher force. But given that the r.d.s. changes, we don't really know a priori where the mechanism changes. This makes the entire extrapolation exercise itself potentially problematic.

R: We appreciate the clarification from the reviewer, as we now understand the comment and the potential source of confusion. We hope we can clarify now. First of all, we agree with the reviewer that according to the measured rate of folding (second timescale), the rate of unfolding in the absence of force should in theory be slower. There is nevertheless one point that potentially we have not made clear enough; in our experiments we first need to unfold the protein (occurring at a relatively low force (~150pN) for a short time (0.3 s) to reveal the disulfide bond to the solution, and then the force is increased in the second pulse (~300-500pN) to accelerate the rupture of the disulfide bond. Hence, in Figure 2d-f of the manuscript we only vary the force of this second pulse (similar to the figure below, from Garcia-Manyes et al *Nat Chem* 2009), and therefore extrapolation of the rate in the absence of force corresponds only to the rupture of the disulfide bond alone, and the previous component of the trace (1st pulse) corresponding to protein unfolding is not taken into account.

We believe that this is important to clarify, because in fact the reduction of the disulfide bond alone exhibits a much shallower dependence with the force than the mere protein unfolding, which in fact gives rise to a seemingly slow unfolding rate constant on the absence of force ($k_0 = 18.5 \times 10^{-4} \text{ s}^{-1}$ (Ainavarapu et al., *Biophysical Journal*, 2007), hence implying an associated average time of ~ 9min. In that sense, both processes

(protein unfolding and disulfide bond reduction) are easily decoupled in the two initial pulses by means of the pulling force. By contrast, in the quench pulse in the absence of force, the kinetics of disulfide reformation is intrinsically coupled to the conformational dynamics of the folding protein. As such, the protein conformation does not play a role in the r.d.s. of disulfide bond scission, and it is not a real concern that the extrapolation of the rate of disulfide reaction is fast in our experiments. We really hope this clarifies.

This discussion made us realise that, while in the main text the explanation of the different force pulses used in our experimental approach was well-detailed, this distinction was not made clear in the caption for Figure 2. We have now clarified by detailing that the grey part of the trace shown in Fig. 2c (15 nm steps) corresponds to the (fast) protein unfolding process, whereas the red part (10 nm steps) corresponds only to the reduction of the disulfide bond.

The fact that it is a bimolecular reaction does not matter. The reaction being extrapolated is bimolecular (and I think the authors mean “second order” and not “first order” kinetics in their response), and so the force-free rate is also the bimolecular rate, even if under pseudo-first order conditions.

R: We agree with the reviewer that the force-free rate corresponds to a (pseudo-first order, since there is always a *single* disulfide bond) bimolecular rate. In fact, triggered by this comment, we have decided not to ‘constrict’ the paper to a bimolecular reaction, and as such we have deleted the ‘bimolecular’ term from the title of the paper.

3. - *On “thermodynamically allowed”: I think my point was misconstrued. It is just that a high energy barrier and positive delta G do not mean that the forward reaction does not occur on a microscopic level. It just means that the rate might be slow, and especially that even if the forward reaction is fast, the reverse reaction is faster. This kind of language is often muddled except when someone is teaching thermodynamics, but again I think it is important that the authors be precise in their language here. Especially given the extrapolated kinetics mentioned in point (2), above, where the apparent force-free forward reaction has a time scale of milliseconds (or seconds, depending on the units in figure 2f, which should be added). To say that a reaction with a measureable (or easily extrapolated but substantial) rate constant is “impossible” or “forbidden” seems misleading, when “slower than the reverse reaction” is precise. Again, this doesn’t make the work any less interesting or important, at least in my view, but seems quite appropriate given the nature of the work*

R: We agree with the reviewer and definitely see their point. We have now strived to avoid the terms *forbidden* and *impossible* throughout the manuscript, and, according to the reviewer’s suggestion, we have specifically included the point on the relative rates of the forward and reverse reactions on page 4 of the revised version of the manuscript. We have also added the units in Figures 2e and 2f.

Especially because the thermodynamics of the reaction are so central to this paper, it would also be useful to be careful about using delta G (which considers the actual position Q of the system) vs. delta G naught (standard free energy of reaction vs. free energy of reaction), which have very different meanings but are mixed in places here that, if taken literally, misrepresent the actual thermodynamics at play.

R: This is an excellent point. Indeed, the DFT calculations are in essence standard-naught-, while the experiments report on the free energy of the reaction. We have now systematically added the 'naught' where fit throughout the text, and also in Figure 1b.

4. - *On “first bimolecular heterolytic uphill etc... reaction”: I don’t think the work is any less interesting or important if it’s not truly the “first” of this type. And I don’t think it is, unless you go on to qualify it further. The 2004 Sijbesma paper with coordination bonds was both heterolytic and thermodynamically uphill. Given it involves platinum complexes, it was likely associative and bimolecular, although the mechanism was not proven. There is stronger evidence in the 2006 Craig paper that the reaction is bimolecular; it is certainly also heterolytic and thermodynamically uphill. In both cases, the bonds were reformed once force was removed, as in this paper. Neither was studied to nearly the extent done here, however, but they absolutely demonstrated that force was necessary to drive an uphill exchange reaction forward (on both kinetic and thermodynamic fronts). This detracts nothing from the current paper!*

R: We honestly did not want to make unreasonable claimings regarding the novelty of our experiments. We completely agree that both the 2004 Sijbesma paper and the 2006 Craig paper triggered thermodynamically uphill reactions, and we have acknowledged this in the introduction of the revised version of the manuscript. However, it is also fair to say that in the 2004 paper, while certainly demonstrating the reversibility of the reaction, force was applied in the bulk with ultrasounds, and as such lacked detail on the energy of the reaction, the followed mechanism or the direction of force application. In the 2006 paper ‘*Single-Molecule Force spectroscopy...*’ we honestly do not see evidence of experiments directly probing the reformation of the bonds when the force was released (perhaps we have overlooked it though). In any case, in these experiments there was not mention of the (thermodynamic) energy, such that kinetics and thermodynamics were not coupled. Be that as it may, we have now toned down our introduction and written the sentence ‘*However, the precise mechanisms by which mechanical force can trigger thermodynamically-uphill reactivity in bimolecular heterolytic systems and how this impacts the reaction reversibility remains largely unknown*’. Similarly, we have also toned down our discussion ‘*However, we lacked quantitative evidence of the mechanisms controlling the reversibility of mechanochemical reactions that are thermodynamically non-spontaneous*’. We hope this sets a clear and truthful reflection of the scope of the present paper.

5. - *Heterolytic: My apologies – I had “heterolytic scission” of the type A-B -> A+ and B- in my head, but the authors are quite right that they do not use that language.*

R: Excellent, we are happy that this was clarified.

This is excellent work, certainly publishable in Nature Commun. once the language and discussion is cleaned up a little further. Because the authors are diving into such rich thermodynamic discussion (which I am delighted to see), I just think that they have put themselves in territory where attention to detail in language is all the more important. No additional experiments are necessary, in my opinion.

R: We thank this thorough reviewer again. Following this reviewer’s suggestion, we have further toned down the language and done our best to keep precise with the thermodynamically-related discussion.

REVIEWERS' COMMENTS:

Reviewer #3 (Remarks to the Author):

I am satisfied with the revised manuscript.